# Algae as Feedstuff for Ruminants: A Focus on Single-Cell Species, Opportunistic Use of Algal By-Products and On-Site Production

**DOI:** 10.3390/microorganisms10122313

**Published:** 2022-11-22

**Authors:** Diogo Fleury Azevedo Costa, Joaquín Miguel Castro-Montoya, Karen Harper, Leigh Trevaskis, Emma L. Jackson, Simon Quigley

**Affiliations:** 1School of Health, Medical and Applied Science, Institute for Future Farming Systems, CQUniversity, Rockhampton, QLD 4701, Australia; 2Faculty of Agricultural Sciences, University of El Salvador, San Salvador 01101, El Salvador; 3School of Agriculture and Food Sciences, The University of Queensland, Gatton, QLD 4343, Australia; 4Coastal Marine Ecosystems Research Centre, School of Health, Medical and Applied Science, CQUniversity, Gladstone, QLD 4680, Australia

**Keywords:** biofilters, cyanobacteria, microalgae, single-cell protein, rumen function

## Abstract

There is a wide range of algae species originating from a variety of freshwater and saltwater habitats. These organisms form nutritional organic products via photosynthesis from simple inorganic substances such as carbon dioxide. Ruminants can utilize the non-protein nitrogen (N) and the cell walls in algae, along with other constituents such as minerals and vitamins. Over recent decades, awareness around climate change has generated new interest into the potential of algae to suppress enteric methane emissions when consumed by ruminants and their potential to sequester atmospheric carbon dioxide. Despite the clear potential benefits, large-scale algae-livestock feedstuff value chains have not been established due to the high cost of production, processing and transport logistics, shelf-life and stability of bioactive compounds and inconsistent responses by animals under controlled experiments. It is unlikely that algal species will become viable ingredients in extensive grazing systems unless the cost of production and practical systems for the processing, transport and feeding are developed. The algae for use in ruminant nutrition may not necessarily require the same rigorous control during the production and processing as would for human consumption and they could be grown in remote areas or in marine environments, minimizing competition with cropping, whilst still generating high value biomass and capturing important amounts of atmospheric carbon. This review will focus on single-cell algal species and the opportunistic use of algal by-products and on-site production.

## 1. Introduction

Algae and cyanobacteria are found in both marine and freshwater environments and are classified as either eukaryotic organisms or prokaryotic cyanobacteria (blue-green algae). Many species of algae have been in the diet of non-Western civilizations for centuries [1] and have been a relatively common feed source in livestock and aquaculture for many years [2]. However, one of the first experiments with single-cell microalgae species as a ruminant feed was conducted less than twenty years ago [3] with larger feeding trials only happening in the past decade with sheep [4] and cattle [5]. Since then, there has been limited adoption in their use in nutrition of ruminant animals mainly due to the high costs involved with algal production and harvesting. Despite this, recent awareness around climate change has renewed interest in the use of macroalgae, e.g., the red seaweed *Asparagopsis* sp., as mitigators of enteric methane emissions from ruminants [6]. To be market viable, algae products require value adding through either carbon reduction funds or marketing linked to the sustainability of beef produced under low emission schemes. Parallel to this, other algae species have been included in ruminant diets to promote good health-related outcomes to consumers of meat [4] and milk products [7]. Again, price is the main challenge for the effective adoption of algae as an orthodox and off the shelf feedstuff. The alternative could come in the form of algae by-products sourced from biotechnological industries. Industries of importance include fuel production [8] and when algae are used as biofilters to remove nutrients or contaminants [9,10] or to ameliorate carbon emission from various factories [11]. The concept of biofilters is of particular interest in extensive grazing systems of northern Australia since it has the potential to be linked to the development of silvopastoral plans for rehabilitating mine sites with the algae produced in wastewater being used as feed for livestock or organic fertilizer during the implantation of trees or to increase pastures productivity. 

In summary, algae can be used in ruminant diets in their raw form if produced in excess, or as by-products from other industries. The differences in nutrient composition cause variable effects within the rumen and animal response. Reviewing the available literature on algae species and products used in ruminant nutrition will provide a greater understanding of the effects on ruminant production and physiology. As algae and algal byproducts become more available at a global scale, this review hopes to recommend new opportunities for efficient in ruminant production systems.

## 2. Species of Algae and Their Nutritional Composition

Algae chemical composition varies from strain to strain and from batch to batch, particularly in protein and lipid content and in the composition of their fatty acids [12,13]. A particular interest has risen around the use of the single-cell microalgae and cyanobacteria as some have the potential to produce more lipids per area than traditional plants used for biofuel production. Another recent interest is focused on bioactive compounds present in red seaweed, such as bromoform, that directly affects methanogenesis. The latter topic will be discussed in greater depth later in Section 6. Nevertheless, the higher cost of cultivating and harvesting the biomass of algal species remains the most critical barrier to market deployment of large-scale commercial production [8]. These microorganisms occur widely in a variety of natural and man-made environments [14] and despite a large number of species identified, i.e., 15,000, only about 15 were in use in industry at the time [15]. Although not currently prevalent, the increased use of microalgae as biofuel has led to the use of the resulting by-product, i.e., post-lipid extraction algae residue (PEAR), as a ruminant feed [16]. The ruminant’s ability to upcycle by-products into high-quality human-edible protein has been highlighted [17] and PEAR retains about 25% of the protein fraction of the original biomass [18]. The latter creates opportunities for inclusion of PEAR in ruminant diets as a protein source, such as with other by-products such as distiller’s grain and soybean meal. A recent study suggested that PEAR could be included up to 60% of dietary dry matter in beef cattle rations [16]. However, the issue with any by-product is that the final nutrient composition is subject to the extraction process, which is highly variable. 

Table 1 presents the nutritional composition of algae species that have been used in ruminant in vivo trials in their raw form only. Some examples include the marine red seaweed *Asparagopsis taxiformis*, the brown algae *Sargassum* spp. and *Macrocystis pyrifera* and the single-cell microalgae *Schizochytrium* spp., *Crypthecodinium cohnii*, *Dunaliella salina* and the freshwater *Arthrospira platensis*. and *Chlorella* spp. 

The mineral composition of algae can be of high relevance depending on a number of factors. The ash content in Table 1 indicates the amount of minerals present in these algal species. However, the mineral profile is expected to greatly change amongst them. A recent paper by Neville et al. [19] demonstrated the benefits of replacing limestone for the calcareous marine algae *Lithothamnion calcareum* as a source of Ca in the diet of dairy cows during the transition period. Mineral-related challenges may be more pronounced in those high producing animals, in which potential differences in bioavailability between mineral sources may play a stronger role. In the latter work, the algae species was added in the diet as a feed additive containing 95% ash composed of 30% Ca.

**Table 1 microorganisms-10-02313-t001:** Composition of macro and micro-algae species used for ruminants in their raw form.

	Protein	Lipids	Ash	NDF	Ruminant Species	References
	(g/kg Dry Matter)					
Macroalgae						
*Asparagopsis* sp.	183	3	504	272	cattle	[6,20]
*Chaetomorpha linum*	103–182	14–20	120–319	319	sheep	[21,22]
*Macrocystis pyrifera*	128	22	386	199	goats	[23]
*Sargassum* sp.	86	6	277	141	cattle	[24]
*Ulva lactuca*	95–211	5–17	175–181	216–415	goats, sheep	[22,25]
Microalgae						
*Arthrospira platensis*	460–744	20–150	47–257	35–87	cattle	[5,7,12,26,27]
*Chlorella pyrenoidosa*	548–600	20–143	64–202	4	cattle	[5,12,28]
*Chlorella vulgaris*	586	123	51	15	cattle	[7]
*Crypthecodinium cohnii*	194	575	69	50	sheep, cattle	[29,30,31]
*Dunaliella salina*	62–570	60–281	90–787	0	cattle	[5,12,28]
*Nannochloropsis gaditana*	385	192	158	219	cattle	[7]
*Nannochloropsis oculata*	289–292	197–292	81–89	69.5	cattle, goats	[32,33]
*Prototheca moriformis*	38–76	81–109	5–70	114	cattle, sheep	[34,35]
*Schizocythrium* sp.	130–208	38–577	74–139	263–369	cattle, sheep	[3,4,36,37]

It is important to emphasise that neither the search for new species nor evaluations of the chemical composition of algae currently in use has reached an end. The biomass of naturally occurring single-cell microalgae species found in northern Australia were evaluated and it was concluded that some protein-rich strains could be used for animal feed [14]. Despite considering the latter information relevant, in this review the authors attempted to focus the discussion on the limited data available from in vivo experiments, relying less on data from in vitro trials or experiments using non-ruminant animal species.

## 3. Effects of Algae on Dry Matter Intake and Apparent Total Tract Digestibility

Ten manuscripts, involving 12 feeding trials reporting dry matter intake were reviewed (Table 2). Five evaluated lactating dairy cattle [7,34,36,37,38] in which the inclusion of *Arthrospira*, *Chlorella*, a mixture of *Chlorella* and *Nannochloropsis* (1:1) [7] and *Prototheca* [35] at levels between 1.72 and 2.50 g/kg BW, had no effect on dry matter intake, or digestibility of dry matter, organic matter, crude protein and neutral detergent fibre. Similarly, both Till et al. [37] and Till et al. [38] observed no effects on dry matter intake with the inclusion of *Schizochytrium limacinum* at levels of between 50 and 150 g/d. Interestingly, a decrease in the concentrate intake was observed when microalgae were included in the diet, but this reduction was compensated by an increase in silage intake [7]. In the latter, the microalgae was included in the concentrate, which could lead to speculation that there was a palatability issue. In this regard, it was reported a clear fishy odour associated with *Nannochloropsis* whereas the smell of *Arthrospira* and *Chlorella* appeared to be more neutral [7]. As stated by the authors, the smell resembling fish might not be common for all algae products, but it is important to keep in mind that the aroma can change with the oxidation of unsaturated fatty acids during processing and storage [39]. Algal processing and storage conditions are variable and therefore odour may differ for each experiment or region.

The only study with dairy cows reporting an effect of microalgae on dry matter intake used *Schizochytrium* fed at 0.22, 0.44 and 0.66 g/kg BW to Holstein cattle and showed a linear decrease in intake [36]. These authors did not discuss the decreased intake, however *Schizochytrium*, the microalgae present in their supplement DHA Gold, is a source rich in fat, which at high levels reduces intake. However, the ether extract of all diets in the study did not vary significantly and remained in the normal ranges for dairy cows. 

It is also important to note that the purpose of the addition of microalgae differed between studies: Microalgae species had been included as source of polyunsaturated fatty acids to modulate rumen fermentation [36,37,38]; added as a source of protein to replace soybean meal [7]; and *Prototheca* (crude protein = 77.9 g/kg dry matter) was added as energy replacing corn [34]. The sample size of microalgae studies on intake and digestibility in dairy cattle is small, and the ration ingredients were diverse between studies. It is therefore difficult to ascertain the effect of microalgae supplementation on intake and digestibility in dairy cows and more studies involving dose responses of different algae and ingredient interactions are required. Nevertheless, providing dietary composition of rations for dairy cows are not novel, microalgae can be included in the diet of dairy cows at inclusion levels of 2.15 to 75.5 g/kg dry matter, without deleterious effects on intake. 

Another five studies involving seven trials have been conducted with either growing cattle or sheep. Costa et al. [5] fed *Arthrospira platensis*, *Chlorella pyrenoidosa* and *Dunaliella salina* (4, 4.7 and 4 g/kg BW, equivalent to 35, 48 and 56 g/kg dry matter) as well as cottonseed meal as a positive control. In this study *Arthrospira* and *Chlorella* supplementation increased the dry matter intake of the basal diet which consisted of a low-quality tropical grass Speargrass (*Heteropogon contortus*) hay. Only *Arthrospira* supplementation increased dry matter intake to the extent of cottonseed meal supplementation. *Dunaliella* supplementation did not increase intake of the basal diet. In this study there were no differences in dry matter digestibility between *Arthrospira*, *Chlorella* and cottonseed meal supplement treatments, while *Dunaliella* showed a similar dry matter digestibility to the control diet. A follow up response trial using either *Arthrospira* or cottonseed meal as supplements [5], found an equivalent quadratic increase in intake with each supplement in steers fed a basal diet of poor-quality hay. An identical linear increase in dry matter digestibility was observed when either *Arthrospira* or the conventional cottonseed meal was supplemented.

The second trial in Costa et al. [5] confirmed previous research of Panjaitan et al. [27] where *Arthrospira* was deposited directly in the rumen of fistulated steers fed a low-quality hay. In this experiment there were quadratic increases in dry matter intake and dry matter digestibility which was associated with an increase in the supply of rumen degradable nitrogen that enhanced microbial activity. It was suggested that the main mode of action of *Arthrospira* was the increase in the protein to energy ratio that led to a higher passage rate in the rumen [27]. This higher passage rate promotes a higher dry matter intake, increases microbial protein flow to the intestine, and likely reduces the maintenance energy requirements of the rumen microbes [41]. When supplemented to steers fed a low-quality basal forage, *Arthrospira* increased dry matter intake and digestibility in a similar fashion to that of other conventional protein meals [5]. 

Another study with growing steers, fed a commercial microalga between 0.44 and 1.56 g/kg BW in a feedlot-type diet (forage to concentrate ratio of 15 to 85) [40]. Dry matter intake linearly increased with increasing microalgae inclusion, but no changes were observed for average daily gain, leading to a decreased feed conversion rate when algae meal was fed. The authors hypothesized that supplement improved palatability, but an increased passage rate may have led to a reduced nutrient utilization and, therefore, a lower feed conversion efficiency. This is supported by an increase in dry matter disappearance rate of microalgae when incubated in situ in the rumen of steers fed increasing algae levels (from 0 to 45%). Another factor that could have contributed to the decreased average daily gain was that the crude protein concentration significantly differed within the study of Van Emon et al. [40] (ranging from 177 to 136 g/kg DM), which may have impaired rumen fermentation and nutrients digestibility, leading to a reduced supply of nutrients for absorption in the duodenum, hence, a lower supply of building blocks for growth. Digestibility, however, was not reported in the latter study.

A study with feedlot diets for sheep reported no effects of *Prototheca moriformis* dietary inclusion on dry matter intake, however a linear reduction of dry matter, organic matter and crude protein digestibility were apparent [35]. Interestingly, both experiments in the latter study found contrasting effects regarding neutral detergent fibre digestibility. In the first experiment, where microalgae replaced soybean hulls, neutral detergent fibre digestibility linearly decreased with increasing algae inclusion, but in the second experiment, where microalgae replaced corn, neutral detergent fibre digestibility increased. As proposed by the authors, these differences relate to the nature of the fibre present in algae, which could be more of soluble nature, but apparently with a digestibility ranking somewhere between that of soybean hulls and corn. Further studies are required to better understand the nature and degradability characteristics of fibre from microalgae.

In general, due to the limited number of studies evaluating the effects of microalgae on dry matter intake and digestibility, as well as the differences in the experimental conditions, e.g., diets, animals, and the microalgae being tested, it is difficult to define the extent of the effects of microalgae supplementation. However, some microalgae can be safely included in diets of dairy cattle, growing steers or sheep as a source of protein or energy without any deleterious effects on intake and digestibility, as long as the basal diet composition remains similar, and for levels of inclusion to be between 2.15 and 75.5 g/kg dry matter. Interestingly, there is evidence that microalgae interact with diet ingredients and affects neutral detergent fibre digestibility. For example, when substituting microalgae for corn, neutral detergent fibre digestibility improves when the concentrate proportion is above 90%. It is known that high starch diets tend to decrease fibre degradation and therefore adding microalgae could potentially increase fibre utilization in feedlot diets

## 4. Effects of Algae on Rumen Parameters

### 4.1. Volatile Fatty Acids

The volatile fatty acids acetic, butyric and propionic are the main short chain fatty acids produced in ruminants through enteric fermentation. These acids are of paramount importance for providing most of the ruminant’s energy supply. However, there is not much information regarding the effects of algae supplementation on their production nor on other short chain fatty acids and the links to microbial protein synthesis in vivo. A negative relationship of *Arthrospira* supplementation level with butyrate proportion, and a positive relationship with proportions of other branched-chain fatty acids has been reported in the work of Panjaitan et al. [27]. Additionally, the latter authors found a quadratic relationship between *Arthrospira* intake and propionate proportions. Subsequently, Costa et al. [5] observed a positive relationship of total volatile fatty acids using the same microalgae. Panjaitan et al. [27] found a positive linear relation of *Arthrospira* with molar proportions of propionate and branched-chain fatty acid, and a negative relationship between *Arthrospira* inclusion and acetate proportion. Interestingly, a quadratic relationship between *Arthrospira* and branched-chain fatty acid reduced in proportion with increasing *Arthrospira* inclusion in the work of Costa et al. [5]. In addition, in the latter work, no differences in total short chain fatty acid concentration, or individual short chain fatty acid proportions between *Arthrospira* and cottonseed meal when supplemented as sources of protein. It was also found by these authors that *Chlorella* and *Dunaliella* both increased the acetate proportions and decreased the branched-chain fatty acid proportions compared to cottonseed meal inclusion. Conversely, Moate et al. [36] found no effects of microalgae inclusion in the diet of dairy cows on total volatile fatty acids concentration or individual short chain fatty acid proportions, except for a linear increase in butyrate with increasing algae inclusion. Importantly, Costa et al. [5] and Panjaitan et al. [27] supplemented microalgae to a basal diet of poor-quality grass, thus changing significantly the composition of the diet consumed, whereas the rations tested in the study of Moate et al. [36] remained similar in composition.

Microbial protein synthesis and rumen ammonia-N both increase in a quadratic fashion with increasing *Arthrospira* inclusion in the diet in Panjaitan et al. [27]. However, no differences were found between *Arthrospira*, *Chlorella* and cottonseed meal in ammonia-N concentration in the rumen nor in microbial protein synthesis between these true protein sources in Costa et al. [5]. Despite this, in the feeding, it was observed a quadratic response of ammonia-N and branched-chain fatty acid proportion to increasing inclusion of *Arthrospira* in the diet in the latter study. Microalgae inclusion in the diet had no effect on the pH of rumen fluid in cattle fed forage based diets [5,27,36].

It is a fact that more research is required to address the effect of microalgae on rumen function. The current information available in the literature does not highlight any obvious negative effect on rumen function in cattle or sheep however, there is not sufficient information to conclude on the possible effects of these algae on the rumen fermentation and the supply of microbial protein post rumen in various feeding conditions and in different ruminant species production systems.

### 4.2. Microbial Synthesis in the Rumen

Suitable conditions, such as absence of oxygen, relatively constant pH, appropriate nutrients, and the absence of growth-preventing inhibitors, facilitate microbial growth in the rumen environment [42]. The main nutrients required for the growth of microbes are fermentable carbohydrates, as source of energy, and N. The energy required for their growth comes from structural or non-structural carbohydrates, depending on diet type. Protein can be also used as energy, but it is usually the most expensive ingredient of the diet [43]. Other nutrients are also required by rumen microbes, such as minerals, e.g., sulfur, phosphorus and magnesium [44] and vitamins [45]. Insufficient amounts of nutrients result in a lower efficiency of microbial protein synthesis in the rumen. Microalgae, besides being a source of both N and energy, are a potential source of these other nutrients for microbes. In addition, Costa et al. [5] indicated that both *Arthrospira* and *Chlorella* were high in phosphorus, often a limiting nutrient in grazing systems across the globe. The most important nutrient supplied by these single-cell microorganisms is likely to be N released on the degradation of algae protein. The extent of degradation of the microalgae within the rumen is not fully known but the lysis and fermentation of microalgae within the rumen may be presumed to follow the normal fermentative process outlined below. A higher efficiency of microbial protein synthesis in steers fed a low protein basal diet supplemented with *Arthrospira* compared to the equivalent rumen degradable N intake supplied by a non-protein N source, i.e., urea, was attributed to the package of nutrients supplied by the microalgae [27]. Despite a higher branched-chain amino acid content in *Arthrospira*, which theoretically could benefit the growth of some bacterial species in the rumen, this microalga achieved the same efficiency of microbial protein synthesis values observed for cottonseed meal, a by-product traditionally used as protein supplement for ruminant animals [5]. 

Microbial protein can provide all the amino acids (AA) required by the host animal, i.e., ruminant animal, but some microbes are able to use pre formed AAs. A spray-dried *Arthrospira* from two different sources had 333 and 361 g/kg of their protein composed of essential AAs [46]. A more recent study reported total AA concentration ranging from 855 to 930 g/kg CP for *Arthrospira*, *Chlorella* and *Nanochloropsis*, including 416 to 464 g/kg CP of essential AA [7]. Therefore, microalgae have the potential also to provide dietary AA for absorption by the animal but the nutritional value will remain contingent on the proportion of this protein that remains undegraded in the rumen (RUP). In this regard, a recent study reported the 48 h in vitro RUP of four species of microalgae: *Arthrospira* (n = 2), *Chlorella* (n = 7), *Nannochloropsis* (n = 7), and *Phaeodactylum* (n = 2) [47]. The RUP ranged between 40 and 61% of RUP of the total protein in non-cell disrupted microalgae, with the highest rumen undegradable protein being found for *Nannochloropsis* and the lowest for *Arthrospira* [47]. These results are in agreement with Costa et al. [5], who found that algal protein has a higher resistance to degradation in the rumen compared to soybean. Importantly, the in vitro intestinal digestibility of the rumen undegradable protein ranged between 270 and 430 g/kg RUP [47], significantly lower than the intestinal digestibility for soybean meal (between 700 and 880 g/kg RUP) and rapeseed meal (between 500 and 820 g/kg RUP) [48]. If the results of these in vitro studies are replicated in vivo, the supply of dietary AA from microalgae would be of a lesser quantity compared to other protein supplements. However, in vitro study results do not necessarily reflect the in vivo rumen degradability of protein and therefore, the subsequent intestinal utilization of the resulting RUP becomes a critical area that deserves further attention. 

Moreover, the combination of protein and the resulting branched-chain amino acids and branched-chain fatty acids, minerals, and vitamins within microalgae contribute as a source of nutrients to microbes and potentially directly to the host. In the work of Panjaitan et al. [27] *Arthrospira platensis* was fed to animals offered a basal diet of low-quality hay, i.e., 3.8% protein, ad libitum. This resulted in a higher efficiency of microbial protein synthesis than animals fed urea, a non-protein N source, with the same basal diet. This indicates that specific nutrients, not just N, influenced rumen microbe activity allowing them to reach microbial protein synthesis up to 550 g/d and efficiency of microbial protein synthesis of 179 g microbial protein/kg digestible organic matter, similar to those parameters reported for cattle grazing temperate grasses [49]. The exclusive use of a non-protein N source results in microbial protein synthesis of approximately 130 g/kg of digestible organic matter [50]. The nutritional attributes of this microalgae with respect to the efficiency of microbial protein synthesis and aspects of its protein degradation in the rumen, need to be further evaluated to better understand the mechanisms by which these microalgae stimulate microbial activity and increase microbial efficiency.

## 5. Fatty Acid Composition 

Lipids sources in the form of fats and oils often depress fibre digestion when present at high concentrations in the diet. This must be accounted for when considering the use of algae since they can be rich sources of lipids, with the most diverse fatty acids profiles. Numerous health benefits can be attributed to specific long chain polyunsaturated fatty acids when present in human diets [51,52,53]. Algae are good sources of polyunsaturated fatty acids such as linoleic and linolenic, both essential for life of all mammals, but they vary in content and composition. When fed as a supplement to animals, algae may alter the fatty acids composition of meat [4,54]. Costa et al. [12] examined the FA profile in the rumen fluid of cattle fed three microalgae species and found that *Chlorella pyrenoidosa* increased polyunsaturated fatty acids concentration in the rumen fluid of fistulated steers, which if transferred to meat, could have health related benefits to consumers. When feeding unsaturated fatty acids to ruminants, the fatty acids profile encountered in the meat will be different to the one present in the diet because of the biohydrogenation process in the rumen [55]. Although some polyunsaturated fatty acids, such as C20:5 n-3, EPA and C22:6 n-3, DHA can escape rumen biohydrogenation [54]. Microalgae that are rich in longer chain polyunsaturated fatty acids with 20 and 22 Cs, include *Schizochytrium* [56,57], and *Crypthecodinium cohnii* [29,58,59]. For humans these long chain polyunsaturated fatty acids are physiologically important, helping retinal and cortical development during early life [52,53]. Pickard et al. [29] fed a marine alga rich in DHA (C 22:6n-3) to pregnant ewes in the final weeks of gestation, i.e., 10 to 6 weeks prior to birth, and reported that lambs from these ewes stood significantly sooner after birth demonstrating an improved vigour compared to the control treatment. 

Long chain polyunsaturated fatty acids are derived from linoleic (C18:2n-6) and alpha-linolenic (C18:3n-3) acids by enzymatic desaturation and chain elongation and cannot be synthesized in the body [53]. The polyunsaturated fatty acids, linoleic (C18:2n-6) and alpha-linolenic (C 18:3n-3) are present in the lipids of microalgae, e.g., *Arthrospira*. However, if these sources reach the rumen they are exposed to transformation by microbial enzymes leading to formation of other free fatty acids, such as C18:3n-3 converted to stearic acid, C18:0, or C18:2n-6, often incompletely biohydrogenated, converting into C18:0 and monounsaturated isomers [54]. 

The longer chain length of C20:5n-3 and C22:6n-3 present in fish oil and some other algae, such as the marine microalgae *Dunaliella* and *Schizochytrium* potentially is the reason for their low rumen biohydrogenation [60]. However, the main fatty acids of *Arthrospira* are the already saturated palmitic acid 16:0 and the polyunsaturated fatty acids C18:1, C18:2 and C18:3 that would be completely hydrogenated to stearic and monoenoic acids in the rumen [61], most likely why Costa et al. [12] did not observe polyunsaturated fatty acids increments with inclusion of this microalgae. 

Dairy and beef are the major sources of conjugated linoleic acid for humans and there are various isomers resulting from rumen biohydrogenation, e.g., *cis*9, *trans*11, or *trans*10, *cis*12) [55]. Sehat et al. [62] found rumenic acid, i.e., *cis*9, *trans*11, to be the predominant isomer (78 to 84%) present in cheese products; however other isomers of conjugated linoleic acid were also identified in small percentages. A portion of these conjugated linoleic acids escape from the rumen and affect lipid metabolism in the mammary gland [55], subcutaneous and intramuscular fat [63]. More importantly, the *trans*10 *cis*12 isomer of conjugated linoleic acid can markedly inhibit fat synthesis in all three tissue types [55,63]. The extent of formation of these inhibitory isomers within the rumen of ruminants supplemented with algae is not well known. More importantly, none of microalgae led to the formation of conjugated linoleic acid isomers known to inhibit fat synthesis [12].

Another issue is that dietary fatty acids of the n-3 family could potentially delay parturition in sheep [64]. Although, Staples et al. [65] suggested an increase in fertility due to effects of linoleic acid and other longer chain fatty acids on the pituitary, ovaries and uterus, rather than from a higher energy status. Nonetheless, algal supplementation improved lamb vigour at birth as a result of the long chain polyunsaturated fatty acids in the algal species *Crypthecodinium* [29]. It is important to highlight that the fatty acids composition of microalgae varies with species and the types of fatty acids have variable effects in which the long chain polyunsaturated fatty acids and n-3 forms are generally associated with positive effects on ruminant production, e.g., newborn vigour, and fatty acids profile of the fat deposited. In addition, no major effects are expected on rumen function from the basal diet in grazing systems [66] nor from the addition of small quantities of lipids to the diet [67,68]. Although, for more substantial inclusions, i.e., above 6% dry matter, fibre degradation can be negatively affected. It is anticipated that further in vivo studies are required to draw more realistic conclusions about algae as a feedstuff for ruminants on this regard. 

## 6. Algae as Enteric Methane Mitigators

Another important parameter associated with the feeding of algae to ruminant animals is the influence on enteric methane emissions. Methane synthesis in the rumen is directly related to the presence of methanogens but also linked to the efficiency of energy utilization which is interrelated with the methanogenic pathway. Methane emissions from livestock account for a considerable proportion of greenhouse gas emissions in the agricultural sector and it is therefore a relevant parameter to study when investigating the potential use of a feedstuff for ruminants. Recently, McCauley et al. [69] reviewed the use of algal-derived feed additives and their influence on enteric methanogenesis in ruminant animals. One of the genera highlighted by the latter authors includes the macroalgae *Asparagopsis*. These red seaweeds contain bioactive compounds, with special emphasis on bromoform that directly affects methanogenesis. Bromoform is a halogenated analogue of methane that inhibits enzymatic activity of methyltransferase by reacting with the reduced vitamin B_12_ cofactor [70]. The latter enzyme is required in methane formation, directly affecting enteric methane emissions. In contrast, the single-cell microalgae species with potential as methane mitigators, such as *Schizochytrium* are high in polyunsaturated fatty acids which work as a H sink, competing with methane formation pathways [3]. Despite being very promising in suppressing methane emissions, production of these algae currently inhibits price competitiveness with other supplements, unless there is value added to beef or milk produced under low emission schemes. Furthermore, as highlighted in a recent meta-analysis evaluating the use of *Asparagopsis*, there was marked heterogeneity in the results of methane reductions [71]. The latter authors found differences in responses which were evident for the red seaweed at the species level. These authors concluded that while there were practical applications to reduce methane emissions, more in vivo experiments are required to strengthen the evidence and to evaluate potential risks for the use of the different seaweed. Both *Asparagopsis taxiformis* and *Asparagopsis armata* have shown to be effective in reducing methane but they contrast in efficacy most likely because of the concentration of bromoform in those species. For example, the concentration of bromoform in *A. armata* was 1.32 mg/g in the work of Roque et al. [6] compared to 6.55 mg/g in *A. taxiformis* in Kinley et al. [20]. Roque et al. [6] observed a reduction of 67.2% at an inclusion rate of 18.3 g/kg dry matter in lactating dairy cows whilst Kinley et al. [20] reported reductions of enteric CH_4_ production of up to 98% at a much lower inclusion, i.e., 3.26 g/kg dry matter, in beef cattle fed a high grain diet. Recently, Glasson et al. [72] discussed some of the benefits and risks involved in the feeding of *Asparagopsis* for the reduction of methane production from ruminants, including the effects it might have on atmospheric chemistry. Compounds recognized as ozone-depleting substances are listed in annexes of the Montreal Protocol and whilst compounds such as methyl bromide and bromochloromethane are listed there, bromoform is not listed as an ozone depleting compound as such. Bromoform itself is classified as a very short-lived substance and therefore has a relatively low ozone depletion potential overall. Bromoform is released slowly during the natural life cycle of algae and during their senescence and decay. It is during this natural transport of volatilized bromoform through the ozone that it will react with radicals and hence undergoes photolysis, which results in the production of water-soluble reactive product gases and inorganic bromine. It is these latter products that contribute to the decomposition of the ozone layer and that are the ozone depleting substances, rather than the compound bromoform itself, and hence why not bromoform, but these other reactive compounds. In the recent work of Glasson et al. [72], the authors concluded that large scale production and use of this red algae as ruminant feed would not negatively impact animal health, food quality, nor cause ozone depletion. 

From the few in vivo studies with cattle focusing on microalgae species, only one peer reviewed article and a report presented results on methane emissions [36,73]. Moate et al. [36] reported no effect on methane in g/d, but a linear increase in g/kg dry matter intake with increasing levels of *Schizochytrium* in DHA-Gold; whilst Klieve et al. [73] indicated reductions on methane emissions on a liveweight and dry matter intake basis by around 22% and 19.4%, respectively. Although, these authors did not clarify which algae species composed the commercial product Algamac. In summary, the interest of studying methane production when feeding microalgae has developed from in vitro studies [74,75,76] which reported reductions in methane concentration in the gas produced from those fermentations. However, in vitro studies utilize only a known amount of feedstuff and therefore do not take into account feed intake, passage rate and average daily gain. To our knowledge, in contrast to recent work done with the marine macroalgae *Asparagopsis*, evidence supporting meaningful enteric methane reductions in vivo using microalgae species is lacking. 

## 7. On-Farm or On-Site Production of Algae Species 

Algae production through mariculture remains a challenge for some species and may limit their distribution to farms located near coastal areas. For example, complex life histories associated with the red algae *Asparagopsis* were restricting the large-scale commercial farming, although significant research funding for this genus has led to recent research breakthroughs with life stage transitioning triggers. Whereas most macroalgae are readily harvested using straightforward and less expensive mechanical methods, it is far more energy intensive to harvest single-cell species that are less than 5 µm in diameter at a concentration of often no greater than 0.5 g/L [77]. The single-cell species are normally so dispersed in nature that they can only be seen when optimal growing conditions generate the blooming of a very dense population of cells. Though formation of blooms is unpredictable under natural conditions [78], controlled cultivations systems (e.g., photobioreactors; open raceway ponds) optimize light, nutrient, temperature conditions to achieve a continual state of bloom [79]. This approach maximizes biomass yield by harvesting microalgae during a continuous exponential growth phase. More control over production variables such as temperature, or the addition of nutrients into the aquaculture system allows greater uniformity of composition of the algae being produced. Some of the most important commercially produced microalgae are the freshwater *Arthrospira* and *Chlorella* [80]. They can all grow on open systems with relatively no contamination by other microorganisms [81]. This is a very important consideration, since harmful algal blooms could potentially compromise the health of animals [82] and affect the microbial community in the water supply [83]. The production of microalgae on-farm could be an important source of protein for cattle especially during the drier months of the year when energy and protein are limiting. *Arthrospira* has been harvested from its natural habitats, such as Lake Texcoco in Mexico and Lake Chad in Central Africa for centuries [46], hence why it was selected for early studies on the potential use of microalgae as a protein source for cattle [5,26,27]. In the work of Costa et al. [5] the freshwater microalga *Arthrospira* was successfully fed to cattle with positive outcomes in liveweight gain, but the authors highlighted the need to devise harvesting methods suitable for those extensive and remote locations that do not require constant maintenance nor high labour input. Beef cattle production systems in northern Australia are predominantly represented by animals raised in extensive grazing systems [84]. Geographical remoteness, large property sizes, and other characteristics of the production systems impose several management challenges on producers [85]. Access to protein-rich supplements and distributing these to livestock grazing tropical grasses with a low protein content presents a challenge within these systems. With this context in mind, microalgae such as *Arthrospira*, *Chlorella*, and *Nannochloropsis* species appear to be the most suitable species for cultivation as protein supplements in situ on remote cattle properties. 

A more opportunistic use of microalgae produced in the semi-arid regions near extensive grazing systems for production of red meat could be linked to recovery and improvement in water quality of pit and recycled mine waters. The main objective of the use of waste water from mines is to avoid having to rely on the scarcely available fresh water within a semi-arid context. The risk, however, is that microalgae adsorb contaminants that could end up in the food chain if fed to ruminants. Therefore, careful attention must be given before use as animal feed since microalgae have the potential to accumulate toxic metals [86]. As previously indicated, these organisms can work as biofilters removing both nutrients and contaminants [9]. Despite this, these organisms could not only be used as feedstuffs but also as organic fertilizers [87] with considerations to avoid accumulation of contaminants into the areas where they are applied. For these options to become reality, a few key elements must be in place which include the production and harvest methods, delivery mechanisms and adequate technical support for producers adopting the technology. In Figure 1, two hypothetical options illustrate the ideas for the use of microalgae as by-products from mine waste water recovery and from on-farm production, where in both the delivered to animals would occur via water systems. It is important to note that the microalgae being produced could either occur on raceway ponds or covered systems, the latter aiming to prevent evaporation.

The main advantage of on-farm production and feeding systems devised on-site is that less-intensive post-harvest technologies would be required and the issues of shelf-life and stability of bioactive compounds would be minimized, considerably decreasing requirement for freeze-drying or new alternative methods of processing such as oil immersions to deliver bioactive compounds [88]. The final algae mix is then offered to animals in existing infrastructure such as water troughs.

## 8. Challenges with Algae

Macro- and microalgae produce different biologically active compounds. A number of those compounds can result in anti-nutritive and occasionally even more profound detrimental effects. There are specific algae toxins, such as domoic acid [89,90] that could result in serious health problems. Other components, such as high iodine concentration was found in some macroalgae [91] that could be toxic if consumed in excess by livestock. Iodine should not be in amounts greater than 8 mg I/kg DM of total diet [92]. These compounds may be beneficial in some circumstances, but if present in excess they may be detrimental to the rumen microorganisms and/or the animal. Recent literature around the use of macroalgae *Asparagopsis* show great promise because of their effect in enteric methane emissions but they highlight the need to evaluate and understand the consequences of bromoform or other halogens transferring into meat or milk. Muizelaar et al. [93] did not detect the accumulation of bromoform in animal tissue but the compound was excreted in urine and milk of dairy cows. The latter is a concern and needs attention soon prior to large scale commercialisation already taking place. 

Freshwater toxic algae have been reported in over 45 countries [94] including Australia, e.g., *Cylindrospermopsis raciborskii* [95]. Through skin contact and ingestion, these toxins can result in many forms of human illness [96]. Therefore, one of the important issues when devising a way of feeding algae on-farm is to closely monitor any potential health issues when feeding animals. Generally, the fresh algae *Arthrospira* has low contamination by other algae and protozoa in open-air cultures [97].

The high nucleic acid content of rapidly growing microorganisms such as microalgae represents a limitation for its use in humans and other monogastrics, due to possible hyperuricemia (presence of high levels of uric acid in the blood) [98]. Ruminants overcome this, as the microbes in their rumen can utilize these nucleic acids. Nucleic acids are totally degraded in the rumen by extracellular bacterial nucleases and then captured and metabolized by ruminal microbes [99].

## 9. Final Considerations

The demand for food and feed is increasing and will so for the foreseeable future. The need to sustain animal production by making a more efficient use of available resources is a major challenge that animal scientists face to prevent the further degradation of land resources. The production of microalgae in close proximity to grazing ruminants and its utilization as feed represents an important opportunity to produce high quality biomass without utilizing arable land. These microalgae exist in a range of environments, and selection and production of naturally occurring, regionally adapted species may be a more robust approach than the production of introduced species. There is a diversity of microalgae species that have been utilized in ruminant diets, and depending on their nature, they could be added to the diets as sources of protein, energy (in the form of fat or in the form of carbohydrates) or other nutrients (e.g., minerals, essential fatty acids) that can enhance the nutritional and health status of the animals and consumers consuming their meat or milk products. 

Despite the small number of studies available, the evidence suggests potential for microalgae to be successfully included in the diets of ruminants. When incorporated in the diet without significant changes in the ration’s nutrients composition, microalgae can be added without substantial changes in intake or digestibility. When supplied as a source of dietary protein, microalgae had similar effects on rumen function and animal liveweight gain to conventional protein sources and better responses than non-protein nitrogen supplements. Microalgae supplementation may reduce the digestibility of nutrients in the diet, particularly at high levels of inclusion, an aspect worthy of further attention. Moreover, some evidence suggests a low intestinal digestibility of rumen undegradable protein with microalgae meal, which is a potentially detrimental feature of these supplements if the protein is lost in faeces. Importantly, not all species appear to cause identical responses on the animal, and their effects are largely related to their nutrient composition. 

## Figures and Tables

**Figure 1 microorganisms-10-02313-f001:**
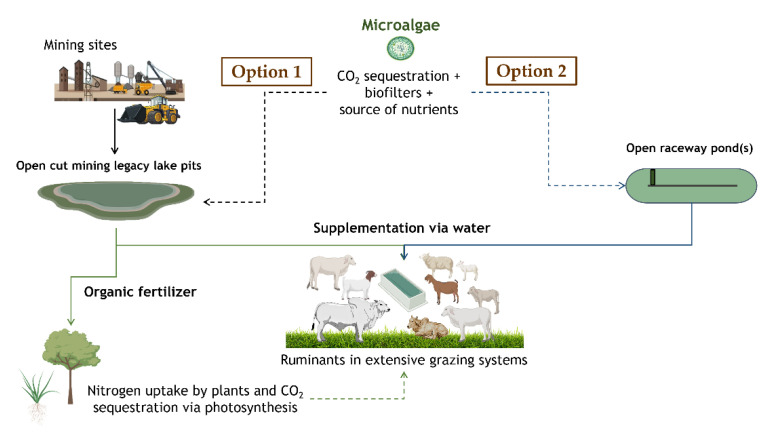
Hypothetical opportunistic on-farm production and water delivery feeding systems.

**Table 2 microorganisms-10-02313-t002:** Summary of in vivo studies testing the effects of algae on dry matter intake (kg/d) and apparent total tract digestibility (g/kg) in ruminants.

Study	Animals ^1^	Basal Diet ^2^	Algae Species	Doses			Diet Composition ^3^			DMI (kg/d) ^4^	Digestibility (g/kg) ^3,4^			
				g/kg DM	kg/d	g/kg BW	CP	EE	NDF		DM	OM	CP	NDF
[7]	Finnish Ayrshire lactating cows (112 + 21. 6 DIM; 36.2 + 3.77 kg/d MY; 652 + 79.5 kg BW)	Grass silage and concentrate. (Forage = 50%)	Control	0	0	0	154	81.1	410	21.5	651	659	617	474
			*Arthrospira platensis*	50.9	1.12	1.72	153	87.3	424	22.0	641	650	602	504
			*Chlorella vulgaris*	64.6	1.35	2.07	154	92.4	409	20.9	650	661	609	491
			*C. vulgaris + Nannochloropsis gaditana* ^5^	75.5	1.63	2.50	150	88.0	421	21.6	651	661	606	516
										ns	ns	ns	ns	ns
[36]	Holstein lactating cows (163 + 9.2 DIM; 20.0 + 3.11 kg/d MY; 571 + 48.1 kg BW)	Alfalfa hay and concentrate (Forage = 75%)	Control	0	0	0	199	30.9	370	22.1				
			*Schizochytrium* DHA-Gold	5.58	0.125	0.22	199	33.6	370	22.4				
			*Schizochytrium* DHA-Gold	11.7	0.25	0.44	200	37.2	366	21.3				
			*Schizochytrium* DHA-Gold	18.3	0.375	0.66	201	28.9	363	20.5				
										Linear decrease				
[34]	Holstein lactating cows (57.7 + 49.4 DIM; 25.3 + 5.3 kg/d MY; 590 + 71 kg BW)	TMR corn silage-based (Forage = 50%)	Control	0	0	0	166	37.6	333	22	737	760	737	668
			*Prototheca moriformis*	52.3	1.18	2.00	163	39.5	345	22.6	736	758	723	666
										ns	ns	ns	ns	ns
[37]	Holstein lactating cows (77 + 17 DIM; 44 + 1.9 kg/d MY; 654 + 42.4 kg BW)	TMR corn silage-based (Forage = 55%)	Control	0	0	0	166		452	23.7				
			*Schizochytrium limacinum*	2.15	0.05	0.076	170		455	23.3				
			*Schizochytrium limacinum*	4.33	0.1	0.153	165		452	23.1				
			*Schizochytrium limacinum*	6.44	0.15	0.229	164		460	23.3				
										ns				
[38]	Holstein lactating cows (22 + 0.5 kg/d MY)	TMR corn silage-based (Forage = 55%)	Control	0	0	0	163		419	22.1				
			*Schizochytrium limacinum*	4.55	0.1	0.153	161		419	22				
										ns				
[5]	*Bos indicus* steers (187 + 7.5 kg BW)	Speargrass (24 g CP/kg DM, 695 g NDF/kg DM) (Forage > 66%)	Control	0	0	0	24	20	695	2.35a	418ab			
			*Arthrospira platensis*	188.7	0.748	4	168	38.8	564	3.96c	455ab			
			*Chlorella pyrenoidosa*	258.2	0.879	4.7	186	52.5	497	3.40b	479b			
			*Dunaliella salina*	52.2	0.131	0.7	35.6	24.6	650	2.51a	412a			
			Cottonseed meal	279.1	1.12	6	172	26.5	537	4.02c	476b			
[5]	*Bos indicus* steers (236 kg BW)	Speargrass (33 g CP/kg DM, 689 g NDF/kg DM) (Forage > 66%)	Control	0	0	0	33	20	689	2.35a	418ab			
			*Arthrospira platensis*	188.7	0.133	0.71				Quadratic increase	Linear increase			
			*Arthrospira platensis*	258.2	0.264	1.41								
			*Arthrospira platensis*	52.2	0.529	2.83								
			*Arthrospira platensis*	279.1	0.79	4.23								
[27]	Brahman-Shorthorn cross steers (250.1 + 10.86 kg BW)	Mitchell grass (38.1 g CP/kg DM; 746 g NDF/kg DM) (Forage > 98%)	Control		0	0				Quadratic increase		Quadratic increase		
			*Arthrospira platensis*		0.125	0.5								
			*Arthrospira platensis*		0.35	1.4								
			*Arthrospira platensis*		0.625	2.5								
			*Arthrospira platensis*		1.525	6.1								
[40]	Steers (292 + 22.4 kg BW)	Wet corn gluten feed + Bromegrass hay (Forage = 15%)	Control	0	0	0	177	21	467	7.19				
			Algae meal	150	1.14	0.44	164	27	450	7.57				
			Algae meal	300	2.53	0.99	150	36	433	8.42				
			Algae meal	450	3.98	1.56	136	43	416	8.85				
										Linear increase				
[35]	Whiteface cross wethers (23.0 + 0.54 kg BW)	Grass hay and concentrate (Forage = 8%)	Control	0	0	0	120	35.9	484	1.31	727	736	602	655
			*Prototheca moriformis*	100	0.114	4.96	122	41.7	442	1.14	721	729	589	613
			*Prototheca moriformis*	200	0.254	11.0	121	37.3	389	1.27	703	710	580	536
			*Prototheca moriformis*	300	0.36	15.7	120	40.8	323	1.2	684	691	572	390
										ns	L	L	L	L
[35]	Whiteface cross wethers (33.7 + 0.55 kg BW)	Grass hay and concentrate (Forage = 10%)	Control	0	0	0	110	28.9	252	1.04	751	764	685	375
			*Prototheca moriformis*	150	0.173	5.12	113	32.9	297	1.15	733	745	670	429
			*Prototheca moriformis*	300	0.387	11.5	110	39.1	330	1.29	698	707	618	447
			*Prototheca moriformis*	450	0.536	15.9	112	43.8	351	1.19	680	689	591	449
			*Prototheca moriformis*	600	0.696	20.7	112	47.6	402	1.16	675	680	593	507
										ns	L	L	L	L

Lowercase letters indicate significant differences between treatments (p < 0.05); ^1^ DIM = days in milk; MY = milk yield; BW = body weight. ^2^ TMR = total mixed ratio; CP = crude protein; DM = dry matter; NDF = neutral detergent fiber. ^3^ CP = crude protein; EE = ether extract; NDF = neutral detergent fiber; DM = dry matter; OM = organic matter. ^4^ The effects of the treatments within a study are portrayed as ns = not significant effect of the algae meal; L = linear effect of algae meal inclusion; superscripts for [5]; “Linear or quadratic increase/decrease” = when a regression equation was reported in the study without specification of treatment means. ^5^
*Chlorella vulgaris* and *Nannochloropsis gaditana* in a 1:1 ratio.

## Data Availability

This is a review of data already published and cited accordingly.

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
