# Peer review of "Algae as Feedstuff for Ruminants: A Focus on Single-Cell Species, Opportunistic Use of Algal By-Products and On-Site Production"

_microorganisms, 2022, doi:10.3390/microorganisms10122313_

Round 1
Reviewer 1 Report (Previous Reviewer 2)
Thank you for addressing the comments of the reviewers.
Author Response
Thank you very much! Your input was greatly appreciated. It allowed us to make important improvements on the final version of this manuscript.
Reviewer 2 Report (Previous Reviewer 3)
The manuscript was signifcantly improved becoming better structured, understandable and the information further detailed. In general all sections were deep revised.The paper is now in condition to be accepted for publication.
Author Response
Thank you very much! We are very pleased with this revised version of the manuscript. Your input was greatly appreciated.
Reviewer 3 Report (Previous Reviewer 4)
Thanks for your revised version. There are a few minor errors as below that need to be fixed.
Title “Algae as feedstuff for ruminants: a focus on single-cell species, opportunistic use of algal by-products and on-site production”. The information on the third part of the current title “on-site production” is missing. Hence, modify the title by deleting this part or add the relevant information in the body of the text.
Both the genus “Schizochythrium” and species “limancinum” need correction throughout the manuscript.
Check the error for “Nannochloropsis gadiatana”
Author Response
Thanks for your revised version. There are a few minor errors as below that need to be fixed.
Title “Algae as feedstuff for ruminants: a focus on single-cell species, opportunistic use of algal by-products and on-site production”. The information on the third part of the current title “on-site production” is missing. Hence, modify the title by deleting this part or add the relevant information in the body of the text.
***Response: Thank you! We appreciate your input. We have discussed the topic on Section 7. We had previously called it “On-farm or localized production”. To avoid confusion, we changed the Section title. ***Line 432 now reads: “On-farm or on-site production of algae species”
Both the genus “Schizochythrium” and species “limancinum” need correction throughout the manuscript.
***Response: Thank you! Corrections done as suggested.
Check the error for “Nannochloropsis gadiatana”.
***Response: Thank you! Correction done as suggested.
This manuscript is a resubmission of an earlier submission. The following is a list of the peer review reports and author responses from that submission.
Round 1
Reviewer 1 Report
The paper under review covers an interesting topic related to the application of algae as feed and feed supplements for ruminants. In general the paper does cover the topic adequately, however there are some shortcomings.
1) The overall approach of the paper is to list and to present in details various published works without providing a rigid conclusion in overall. There are plenty of points that are not interrelated and there is a problem with the continuity of the sentences. The paper needs an essential revision and rewriting.
2) the most problematic part is section 3. The whole section 3 is very bad written and contains numerous errors and in general is unclear. There is a confusing on how Spirulina term is used. Please clarify if it is Arthrospira platensis or Spirulina spp. (the latter is a toxic cyanobacterium).
lines 96-99 are not clear. Please rephrase
lines 102-104. It is not clear what the scientific importance of this point is. Please remove this sentence or rephrase.
Lines 102-109. The whole piece is not clear. What is the main point here discussed ?
lines 109. What kind of stressors? This is not clear at all.
3) minor comments: please explain the various abbreviations in the manuscript. The text was not easy to be read.
Please do not give page numbers in the text but in the reference section
In overall I recommend that the manuscript should be rewritten and resubmitted
Reviewer 2 Report
This is a well-written, timely paper that describes most of the work with microalgae and ruminant nutrition. My comments are only places to consider strengthening the discussion.
Abstract--lines 20-24. In some countries there may be stringent regulations prior to use of algae as a feed additive. Suggest this sentence be softened to not say "would not require rigorous control".
In section 2 add a discussion on the bromoform variabilty or reference the discussion later in the paper.
Section 3. This might be the most important section. Please consider adding to this section specifically about how to alter microalgae content to improve the products uniformity of nitrogen, lipid, minerals and bromoform.
In section 4 discussion of the rate and extent of nitrogenous compound degradation could be added or refer to the ruminal ammonia comments later in the paper. Additionally, lipid digestion might be introduced here.
Section 5. Typo at line 220
Section 8 high lipid content can depress fiber digestion. Maybe add a sentence or two relating the fiber digestion data and the lipid content,
Section 9 The presence of bromoform or other halogens in meat or milk should be discussed as a research need.
Reviewer 3 Report
The review focus an interesting topic about the application of microalgae and cyanobacteria as feedstuff to ruminants highlighting the benefits and the processes involved that contribute to a health and sustainable production and high biomass. Explicit information about sustainable prodution and the healthy contribution to human-beings should be pleasant to be better referred in the manuscript.The structure of the manuscript should be revised to not change and come to the same topics several times, contributing for a smooth way of presenting the information. There are some topics and information that need to be better explained and further detailed. Specific comments are given in the document in attachment to help authors to improve the manuscript.

Reviewer 4 Report
Review comments on “Algae and cyanobacteria as feedstuff for ruminants: a focus on 2 single-cell species”
General comments:
This review article emphasizes the potential applications of cheaply produced microalgal biomass as a source for ruminant feedstuff with anti-methanogenic properties, which is very important from global environmental perspectives (CO2 capture as well as GHG emission reduction). However, this review needs some improvements.
To be more focussed, the title can suitably be modified as “Algae as feedstuff for ruminants: a focus on single-cell species like microalgae and cyanobacteria”.
Please refer to the following articles and improve the content with appropriate discussion
Effects of Microalgae Species on In Vitro Rumen Fermentation Pattern and Methane Production November 2019 DOI:10.2478/aoas-2019-0061
In vitro ruminal fermentation and methane inhibitory effect of three species of microalgae https://doi.org/10.1139/cjas-2019-0187
Effects of Chlorella vulgaris, Nannochloropsis oceanica and Tetraselmis sp. supplementation levels on in vitro rumen fermentation https://doi.org/10.1016/j.algal.2021.102284
Chlorella vulgaris microalgae in ruminant nutrition: a review of the chemical composition and nutritive value DOI: 10.2478/aoas-2020-0117
Specific comments:
Keywords: Replace “Blue green algae” with “Cyanobacteria”. “Micro-algae” should be “Microalgae”.
At the beginning of the introduction section, add a sentence or two describing algae in general and cyanobacteria and microalgae in particular, and then emphasize the later group in the context of this review.
Line31: Remove page numbers throughout after the reference (unless it is not objectionable by the Journal)
Line62: Replace “microorganisms” with “organisms” as many algae are not microorganisms.
Line350: Typo “Schizocythrium”
Line338: “7. Algae as enteric methane mitigators”: Add more information in this section from the suggested references above with exapmples from microalgae.
Line380: “8. Fatty acid composition of algae and effects on ruminant animals” It will be helpful if a table is added in this section comparing (same as that of Table 1) the standard fatty acids profiles content of the most potential microalgae that have shown anti-methanogenic properties.